# Wisdom of the Crowds for Forecasting: Future Event Prediction Based on Summarizing Text-based Forecasts

## Abstract

Future Event Prediction (FEP) is an essential activity whose demand and application ranges across domains. While methods like simulations, predictive and time-series forecasting have demonstrated promising outcomes, their application in forecasting complex global events is not entirely reliable owing to the inability of numerical data to capture the semantic information of events. One effective way is to gather collective opinions on the future to make effective forecasts as cumulative perspectives carry the potential to evaluate probability of upcoming events, or at least they can serve as a complement to other approaches. We organize the existing research and frameworks that aim to support Future Event Prediction based on Crowd Wisdom (FEP-CW) through aggregating individual forecasts, the challenges involved, available datasets, as well as the scope of improvement and future research directions in this domain.

## 1 Introduction

The ability to correctly predict the future is of utmost importance to foresee and plan for likely outcomes. Not only does it allow people to take necessary measures, but also concerned authorities can formulate required policies and make informed decisions. For example, a company's business strategy and profits depend largely on how competent they are at predicting future trends. Examples include forecasting the weather (Abhishek et al., 2012; Salman et al., 2015; Kothapalli & Totad, 2017; Singh et al., 2019), stock-market trends (Carta et al., 2021; Park et al., 2022; Nabipour et al., 2020; Pang et al., 2020), geo-political unrests (Goldstone et al., 2010; Mellers et al., 2023; Nebbione et al., 2019), resource crises (Niemira & Saaty, 2004; Parolin et al., 2020; Sakaki et al., 2010) or a pandemic outbreak (Hall et al., 2007; Panagopoulos et al., 2021; Chauhan et al., 2021; Lucia et al., 2020; Ma et al., 2022) like COVID-19 has proven to be of great social and economic importance over the years. However, understandably accurately forecasting the future is an extremely challenging task owing to its inherent uncertainty. While the reasoning capability and the acquired experience give humans an edge over machines when it comes to forecasting likely events, the vast amount of potentially relevant information requires computational approaches. With the rise in artificial intelligence, automated forecasting systems with reasoning abilities matching those of humans or beyond are not unimaginable.

There are many ways in which future forecasting could be realized. For example, time series-based prediction is commonly done in weather forecasting, stock markets trend analysis like options trading and in pricing. Researchers also use simulation techniques like Monte Carlo (Mooney, 1997) to model a sequence of events occurring in time. These particular approaches essentially rely on the history of a phenomenon such as past records of stock prices, previous weather conditions over time, etc. Yet, to predict the future, humans often tend to resort to the opinions of others, such as friends, relatives or professionals to leverage collective wisdom and compare diverse perspectives. This approach takes advantage of the contextual understanding ability of humans that may not be captured by purely data-driven approaches. "Wisdom of the Crowd" in the forecasting context refers to the opinions and expectations that people share about a certain future event or state. Our survey outlines the computational approaches aiming to harness the wisdom of the crowd for future prediction.

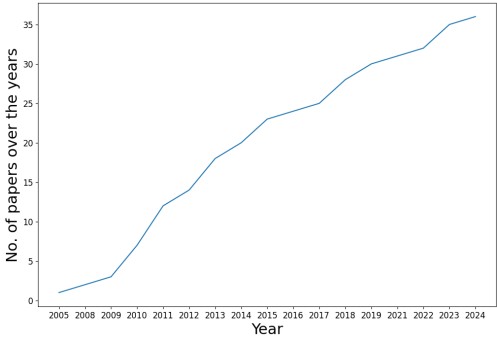
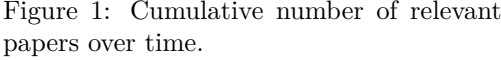

Figure 1: Cumulative number of relevant papers over time.

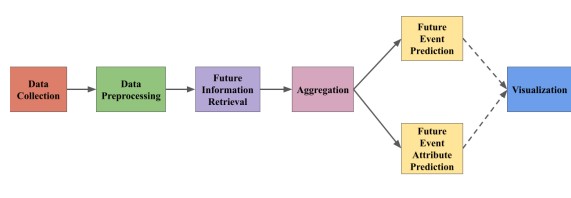

Figure 2: General Flow of Predicting Future Event

To the best of our knowledge, this is the first survey on this topic, despite the importance and the inherent complexity of the task. Our focus is to provide a comprehensive deep-dive into the underlying concepts and techniques developed to target the problem of FEP-CW based on aggregation of multiple individual forecasts in order to facilitate better research contributions in this domain. We overview the concept of FEP, the type, method and sources of data collected for this specific task, the various approaches undertaken by researchers to extract temporal information from this data, methods of retrieving specifically future-related information and finally the techniques applied to predict a future event or future event attributes. Additionally, we also provide a data model of a future-related expression in order to single out its elements that are particularly useful for FEP-CW based on aggregation of multiple individual forecasts.

## 1.1 Related Surveys and Areas

As for the related works, surveys like (Gmati et al., 2019) and (Zhao, 2021) overview more mathematically oriented methods of event, time, and location forecasting. There also exist surveys that are either specific application-based prediction surveys like weather (Cloke & Pappenberger, 2009; Doswell III et al., 1993), civil unrest (Ramakrishnan et al., 2014) or ones that utilize data from a particular domain like Twitter for analyzing whether sentiments extracted from tweets improve forecasts of social, commercial or economic indicators (Arias et al., 2014), the pros and cons of social media for event forecasting (Phillips et al., 2017) and the realms of future human-related events that can be predicted from the opinions expressed in social media platforms (Yu & Kak, 2012). However, none of those works approach reviewing the methods of harnessing the knowledge of the crowd. We also put forward the proposal for a data model of an individual future-related statement and single out the components that are crucial to predicting a future event. So far, none of the works in this area have highlighted the utility of such a data model.

FEP-CW shares similarities with Multi-Document Summarization (MDS) (Cui & Hu, 2021; Zhou et al., 2021; Christensen et al., 2013; Celikyilmaz & Hakkani-Tur, 2010), while having some key differences. In FEP-CW, the goal is to *gather future-related statements* from multiple sources, *aggregate them*, and *obtain a concise summary of possible future events*, a process akin to MDS. However, FEP-CW also entails *conflicting expert opinions and viewpoints*. Resolving these divergent views is an extra step over MDS which assures relevant predictions. While understanding the text semantics is of utmost importance in both tasks, FEP-CW additionally requires *temporal interpretation of information* to make credible forecasts. On top of that, it places more stress on leveraging recent information, unlike MDS, whose aim is to aggregate the majority of the information available into a concise snippet. Lastly, while MDS is solely based on the available evidence, FEP-CW is characterized by *varying levels of uncertainty associated with individual predictions*. Some predictions might be backed by direct implications of available information, others might be more speculative in nature, lacking the sureness quotient.

### 1.2 Publication Collection

We searched for papers relevant to the topic on Google Scholar, DBLP, ACM Digital Library, IEEE Digital Library, Arxiv, and similar digital libraries. Following is an outline of the steps we followed:

1. **Base Search**: We ran multiple searches using queries like *"Future event forecasting", "Future event prediction", "Event forecasting", "Event prediction", "Future retrieval", "Temporal expressions", "Extracting future events", "Future-related information", "Investigation of future events", "Future reference extraction"*, and so on.

2. **Backward Snowballing**: Once we obtained a handful of papers (referred to as the *start set*) through the initial search, we analyzed those works to find other additional related papers from the references and citations of the base set of papers.

3. **Forward Snowballing**: To increase our collection, we identified works that had cited the base set of papers, mostly with Google Scholar's features.

4. **Final filtering**: Our aim was to build a collection of papers on the wisdom of the crowd concept for FEP. We made sure that papers based on other methods like time-series forecasting, simulations, etc, were filtered out from the final list.

In total, 36 relevant papers were selected. We show their cumulative distribution over time in Figure 1 which suggests that this research topic has been steadily investigated over the past years.

An event can be defined as a condition, fulfilling three key characteristics: it happens in the real world, is associated with a geographical location and a temporal specification (Das et al., 2017). In the past, humans have tried to anticipate the occurrence of events in advance through a variety of ways. Starting from ancient times when predictions were based on trajectories of celestial bodies or astrology, the approaches have slowly progressed to more refined methods like scientific modeling and advanced computational algorithms. In science, there is even a field of studies called Futurology[1] or future science that aims to study past and present trends for forecasting future scenarios. The term "Futurology" was coined by Ossip K. Flechtheim, a German professor with the vision of a new probabilistic branch of knowledge. The interdisciplinary nature of the subject arises from its application across diverse domains including business, economics, technology, climate and the society at large. Figure 2 briefly outlines the basic steps involved in future event prediction, which will be discussed in detail in different sections of the paper.

## 2 Data and Sources

Several datasets have been created and used till now. For example, databases like *Sigma Scan*[2] consist of future information from over 2,000 document sources and *Future Timeline*[3] dataset contains more than 13,000 future predictions collected from news articles, research papers, and government reports. However, a major drawback of these datasets is them being time-sensitive. It is impossible to keep them up-to-date which hinders reproducibility. Questions about future events become useless after the resolution dates of these events, as the outcomes become known on the Web or within large language models (LLMs) trained afterwards. A related challenge lies in the fact that accumulating future-related content should be considered as part of the task such as in the current working systems like Regev et al. (2024).

Most of the generic FEP-CW methodologies and analysis papers leverage news articles, tweets and general information about past, present or future events available over the Web. These are further filtered on attributes like topic, event timeline, presence or absence of explicit temporal expressions, etc. While some works have built a refined dataset of their own (discussed in the next paragraphs), others like (Radinsky et al., 2012; Nakajima et al., 2014; Radinsky & Horvitz, 2013; Radinsky et al., 2008; Nakajima et al., 2020) just retrieve Web content to suit their use case. Table 1 outlines the metadata of the major datasets that use the wisdom of the crowd for future event prediction.

---

[1] https://www.britannica.com/topic/futurology
[2] http://www.sigmascan.org/
[3] http://seikatsusoken.jp/futuretimeline/

Table 1: The datasets used in crowdsource-based future event prediction research.

| Dataset | Language(s) | Contents | Count |
|---|---|---|---|
| Jatowt & Au Yeung (2011) | English | News | 3.6M |
| Dias et al. (2011) | English | Web contents | 508 |
| Jatowt et al. (2013) | English (EN), Japanese (JP), Polish (PL) | Web contents, News | 1.04M (EN Web), 2.99M (JP Web), 714K (PL Web), 196K (EN News) |
| Hürriyetoğlu et al. (2013) | Dutch | Twitter | 94,285 |
| Hu et al. (2017) | Chinese | News | 155,358 |
| Goyal & Durrett (2019) | English | News | 5,000 |
| **FORECASTQA** Jin et al. (2020) | English | News | 10,392 |
| **Autocast** Zou et al. (2022) | English | Articles on Economy, Politics, Science, Social, and Others from forecasting tournaments. | 6,707 |
| **IntervalQA** Zou et al. (2022) | English | Generic articles | 30K |
| **ExpTime** Yuan et al. (2023) | English | Compilation of event forecasting datasets | 26K |
| Regev et al. (2024) | English | News | 6,800 |

The majority of the works that perform FEP-CW opt for news articles as their data source, likely due to their vast availability and up-to-date information about events happening around the world. An investigation by Kanhabua et al. (2011) revealed that about one-third of all sentences in news articles consist of some kind of reference to the future. The amount of information available across various time frames help in training a predictive model on past details and validating its predictions for an event for a later time stamp, whose updated status has been already published in the news. In the FORECASTQA dataset (Jin et al., 2020), news articles are collected from LexisNexis[4], filtered for non-English texts from 2015 to 2019 and converted to *<Question, Answer, Timestamp>* triples which could be used to answer binary and multiple-choice forecasting questions. However, FORECASTQA suffers from ambiguity and the lack of context, owing to it being crowdsourced. The unstructured nature of this dataset requires the model to additionally link relevant events to answer forecasting questions. Addressing these challenges, Zou et al. (2022) released the Autocast (Collection of forecasting questions of type True/False, Multiple-Choice or Numerical) and IntervalQA (Large collection of questions having numerical answers only) datasets. Goyal & Durrett (2019) builds a dataset containing temporally ordered event pairs - *<sentence1> <sentence2> <event1> <event2> <temporal relation>*, with *Gigaword* (Graff et al., 2007) serving as the underlying source of articles. The TimeLlaMA (Yuan et al., 2023) model for explainable event prediction is based on a multi-source instruction tuning dataset, ExpTime (Yuan et al., 2023) with documents, forecasting questions, predictions and explanations for these predictions. Regev et al. (2024) built a dataset containing 6,800 manually labeled future and non-future related sentences based on sentences obtained from sources like *Longbets*[5], *Horizons*[6], *ChatGPT*[7] and *New York Times*[8] articles. In (Jatowt & Au Yeung, 2011), news articles are fetched from *Google News Archive*[9] using queries belonging to categories like countries, companies, persons, and others.

Some works like (Dias et al., 2011), (Jatowt et al., 2010) and (Jatowt et al., 2009) choose to focus on the Web, collecting snippets containing temporal expressions and future mentions, rather than limiting information

---

[4]https://risk.lexisnexis.com/

[5]https://longbets.org/

[6]https://radar.envisioning.io/horizons/

[7]https://chat.openai.com/

[8]https://www.nytimes.com/

[9]http://news.google.com/archivesearch

from just news and tweets. For example, Radinsky et al. (2008) mine Google Trends[10] to gather training data which includes the history of 5 years of user search queries.

While most of the works focus on English news or tweets, few tasks have been accomplished on datasets in other languages like Chinese, Japanese, Dutch, Polish, etc. Hu et al. (2017) crawls a Chinese news event dataset containing 15,254 news series from Sina News[11] while Jatowt et al. (2015) collects both English and Japanese tweets with explicit temporal references, structuring them as *<Users, Time-referring messages, Subjects>*. Hürriyetoğlu et al. (2013) and Tops et al. (2013) specifically harvested tweets from a Dutch database.[12] Datasets like (Jatowt et al., 2013) contain both English and non-English (Japanese and Polish) contents queried from Bing[13] web and news search.

Prior to retrieving future-related information, a few general text-preprocessing steps are typically carried out using commonly available toolkits like OpenNLP[14] for sentence splitting, tokenization, part-of-speech tagging, and shallow parsing, TARSQI Toolkit[15] for annotating documents with TimeML[16], SuperSense[17] for named-entity recognition, etc. Documents retrieved from the Web are also filtered out for duplicate URLs.

## 3 Extracting future-related information

Information retrieval involves retrieving documents or text from documents and databases which can be achieved by supplying user queries, specific questions, particular dates or a timeline, etc. Baeza-Yates (2005) was one of the first to introduce the concept of a search engine for extracting future temporal expressions from news articles. They investigated almost a half million future referring sentences from Google News and inferred that the closer the forecasted event, the higher the confidence and accuracy of prediction.

Sentences referring to the future are the most useful in forecasting the unfolding of future events. They provide the relevant context and background knowledge about a future event in question. Below is an excellent example by Nakajima et al. (2018) of how future-referring sentences support answering questions about the future.

Question: Predict whether in 2020 nursing care fee will be applying AI in Care Planning.

Options:

1. AI will be applied in Care Planning.

2. The situation will not change.

3. Care Planning will be created using new method other than AI.

Example relevant sentences that refer to future:

1. The government plans to revise the nursing care fee in future.

2. The government announced that it would develop the legal system and rules for society with AI.

3. A model in which people and artificial intelligence work together will be constructed.

When the publication time of an article is known, it is comparatively easier to find future-related information based on explicit dates. Sentences in articles containing dates or references later to the publication date of the

---

[10]https://trends.google.com/trends/
[11]https://news.sina.cn/
[12]https://ifarm.nl/erikt/twinl/twiqs.html
[13]http://www.bing.com/
[14]http://opennlp.sourceforge.net/
[15]http://www.timeml.org/site/tarsqi/
[16]http://www.timeml.org/site/index.html
[17]http://sourceforge.net/projects/supersensetag/

articles could be identified as future-referring instances if their embedded temporal expressions are referring to the time after their publication dates; although it may not be the case if they are still future-related at a certain time of search. However, the absence of explicit mention of any date or time makes it difficult to retrieve future mentions. One might look for words like *"will"* or *"future"*, but noisy phrases like *"...plans to"* or *"...hope to"* might exist as well. In general, there can be many different ways one can refer to the future, including the use of tenses other than the future tense.

## 3.1 Based on Time

To determine if an event will happen in the future, the first step involves extracting temporal expressions from the data. Often there is a high connection between the reliability of an event and its temporal distance. A temporal expression can be defined as a sequence of words that denote a point in time, a time interval or the frequency of occurrence of an event. To retrieve time-scoped information from unstructured text, it is important to detect, classify and normalize time expressions. Temporal expressions can be explicit, relative, or implicit. Explicit or absolute temporal expressions directly indicate a time point or an interval, for example, 1st April 2024". Relative temporal expressions refer indirectly to a point of time, by means of expressions like *"two weeks later"*, *"a month ago"*, etc. Implicit temporal expressions refer to events like, "The Asian Games" that signify a time event. Besides, temporal expressions can have different granularity levels like hours, days, months, seasons, years, etc. Detecting and extracting them is an important step in FEP-CW as it allows us to differentiate between time stamps, segregate future events from past events, allow chronological ordering of events and understand causality between events. However, owing to the varied nature of words indicating past, present or future, extracting time expressions is not trivial.

### 3.1.1 Time-referenced events

This section deals with future event extraction methods which are structured around the concept of time, like dates, hours, months, years, etc. We discuss methods as early as temporal taggers to the latest time expressions proposed by researchers.

Mani et al. (2005) built the TimeML framework through meticulous annotations of temporal events, marking the commencement of heavy utilization of text data for understanding the underlying relation between time and events. Build on top of the TimeML framework are commonly used temporal taggers like GUTime (Mani & Wilson, 2000), (Strötgen et al., 2010), Stanford CoreNLP tagger [18] and SUTime[19]. However, these taggers possess some limitations. For example, Regev et al. (2024) reported that SUTime failed to perform in ambiguous cases. For example, it mapped *"On Tuesday"* to the upcoming Tuesday, rather than a past Tuesday referred to in the sentence *"On Tuesday, he revealed his decision, which will become official soon."*.

To tackle such intricacies, Jatowt et al. (2010) analyzed time-referenced future-related information by crawling data through structured queries containing temporal expressions defined as $temp_{modifier} + (the) year(s) + yyyy$ (where, $temp_{modifier}$ is a preposition) for retrieval by years, which translates to expressions like *"in year yyyy"*, *"in the year yyyy"*, *"by the years yyyy"* and so on. Similarly, for retrieval by months, the expression looked like *"month_reference + yyyy"* where $month\_reference$ is a full or shortened form of the name of a month (like, February or Feb). ChronoSeeker (Kawai et al., 2010), on the other hand, retrieved both past and future time-referenced events that are relevant to the user's query and filtered noisy events using support vector machines. To analyze the impact of temporal features on the classification of different genres of Web snippets, Dias et al. (2011) employs a pattern-matching methodology as proposed in (Campos et al., 2011) which particularly focuses on years. The unavailability of a Japanese tagger led Manning et al. (2014) to build their own tagger by matching tokens in the tweet content with a dictionary of lexical expressions related to time in Japanese. A set of regular expressions was also defined to capture mentions of hours, days, weekdays, months, seasons, national holidays, and years.

---

[18]https://stanfordnlp.github.io/CoreNLP/
[19]https://nlp.stanford.edu/software/sutime.shtml

### 3.1.2 Time-unreferenced events

Time-unreferenced events do not contain any explicit reference to time and there could be numerous ways to refer to the future. This is also known to be the most common scenario in news collections. Below are some of the methods developed to query future events without explicit time reference.

Searching with keywords like *"will"* or *"future"* might surely yield some results but has a high risk of returning more false-positive cases. For example, a sentence like *"He has a strong will to pursue higher studies"* might be misleading. Kanazawa et al. (2011) proposes a technique with the intuition that the topics of a future sentence should be discussed in future documents and similarly for past-referring sentences. To classify a future-referring sentence in a large corpus, they detect characteristic terms that frequently appear in either the past or future collection. The sentences are ranked based on the average weight of characteristic terms in a sentence, which if contains many highly-scored future characteristic terms and few highly-scored past characteristic terms are more likely to refer to a future event. A better way than statistical counting is to use trained neural network classifiers like DistilRoBERTa [20] as in (Regev et al., 2024). However, the problem lies in the fact that this approach cannot be used to search for future-related content, but rather as a post-processing tool to filter non-future-related content.

## 3.2 Based on Patterns

The complexity of natural language might often mislead a model to perceive future information as past reference. Consider the sentence, *"Carol has decided to open a new bakery"*. Despite the use of past tense like *"decided"*, the sentence indicates an upcoming event. To mine such complex future information, sentences were represented in a morphosemantic structure (Levin & Hovav, 2017) which is a combination of semantic role labeling and morphological information and can extract future reference sentences containing both explicit and critical cases of implicit expressions and context-dependent information.

Nakajima (2015) investigates patterns in multiple news corpora, that are semantically and grammatically consistent, but lexically different. They propose a method for automatically extracting such frequent patterns and establish its effectiveness in the downstream task of identifying future-related sentences. Nakajima et al. (2020) use frequent combinations of such patterns for classifying future-referring sentences by using an analyzer MeCab[21], for extracting morphological information for Japanese words. Ni et al. (2015) and Al-Hajj & Sabra (2018) apply a similar approach to analyze future-referring expressions in English and Arabic. Nakajima et al. (2014) prefer a sentence extraction system *SPEC* based on semantic role annotations for extracting frequent sentence patterns for a corpus. The approach is based on the intuition that a variety of future expressions exists and sometimes few words or phrases are used as future expressions only in a specific context. Hence, there must exist characteristic patterns (grammatic or semantic) appearing in future-related sentences that distinguish it from non-future referring sentences.

## 3.3 Based on LLMs

LLMs have mastered themselves in natural language understanding (contextual, semantic, etc.) and generation. Since LLMs are trained on huge text corpora, they do not need to explicitly collect any data as in all the previous approaches. This information aids an LLM to capture a variety of opinions including those on the future from news articles, web pages and social media, on any specified topic provided by the user. In addition, LLMs can be used as a summarization tool for scrapped web pages. In all of these cases, special prompts need to be designed to elucidate information about the future. A study by Yuan et al. (2023) shows that directly prompting ChatGPT (Ouyang et al., 2022) to identify future-related content leads to suboptimal accuracy. Hence, LLMs provide innumerable avenues for future event extraction.

---

[20]https://huggingface.co/distilbert/distilroberta-base
[21]http://taku910.github.io/mecab/

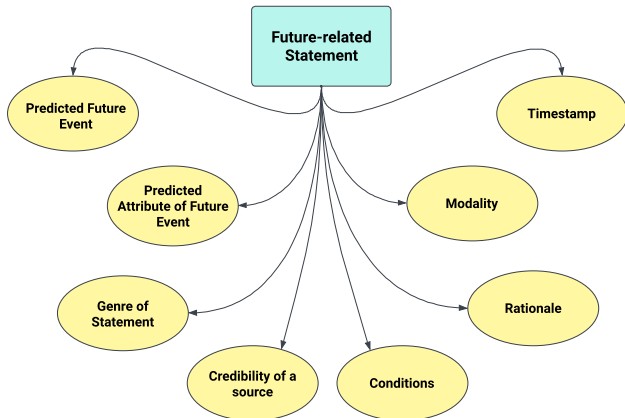

Figure 3: Atomic Components of Individual Future Prediction that can be harnessed for FEP-CW.

## 4    Predicting the Future

Often, humans tend to analyze past events, aggregate insights, and make a highly probable assumption about the possibility of a future event. The future can be predicted by either analyzing the current situation and the latest trends or by examining and extrapolating the past, usually by finding previous occurrences similar to the present ones. Determining probable global events in politics, economics, and business requires extensive common sense and knowledge, which are possessed by human experts belonging to the related domain. To predict a future event, a model must possess the above capabilities, understand the present state and reason about the future state. However, this approach might be quite challenging due to difficulty in collecting relevant data, potential misinformation, lack of understanding of the true cause driving an event, diversity in opinions about how the future could be expressed, conflicts between competing forecasts, amount of uncertainty expressed in forecasts, provided conditions for events to happen, lack of explanations or relevant dates of events or absence of any patterns in past information archives. A majority of works formulate the problem of event forecasting as a problem of link prediction and temporal knowledge graph prediction (Ma et al., 2023a; Lee et al., 2023; Ma et al., 2023b; Deng et al., 2020). Any future-related information has some level of uncertainty. It has been evident that recent documents contain more dependable future-related information than older documents, thus making the younger document an obvious choice in event of conflicts. Despite these challenges, more refined methods are being developed to cater to the needs of the hour. The event prediction methodologies mainly vary due to the underlying heterogeneity of the prediction output. The output might range from the prediction of a whole event to attributes of an event like date, location, time, etc. Additionally, the type of output might be a sentence, a collection of sentences, a word, True or False, etc. depending on the goal of the paper.

### 4.1    Data Model of Future-related Statements

Before we proceed with discussing approaches for FEP-CW based on aggregating individual forecasts, we first outline our proposal of a data model that decomposes a future-related opinion into several components. These components can be useful during the aggregation or summarization process for generating final predictions.

Figure 3 shows the component breakdown of a future-related statement into components that can play key roles in FEP-CW. We describe its constituents below:

- **Predicted Future Event** refers to an event that is expected to occur by the statement creator.

Table 2: Example of the proposed data model components

| Context: (July 31, 2024) Japanese National Bank will try to increase the value of Yen after October since the country needs to import many goods and their current price is too high. This however would depend on the actions of FED and the changes in the interest rate in the USA. | |
|---|---|
| **Component** | **Example** |
| Predicted Future Event | Increase the value of Yen. |
| Predicted Attribute of Future Event | Date: After October 2024. |
| Genre of Statement | News |
| Conditions | Actions of FED and the changes in the interest rate in the USA. |
| Rationale | The country needs to import many goods and their current price is too high. |
| Timestamp | July 31, 2024 |

- **Predicted Attribute of Future Event** are any associated information about the future event that the statement creator forecasts in her future-related statement (e.g., its actors, location). Of particular interest is here the information about the date and time of the forecasted event.

- **Genre of Statement** defines the type of document in which the statement appears setting up the general context for understanding the statement. News, web and social media are a few of the common sources.

- **Credibility of a Source** determines the quality and reliability of the statement creator (a person such as an expert or average user, an institution, etc.) which helps to assess the quality of the target future prediction.

- **Conditions** are usually some kind of specific circumstances or constraints that need to occur for the event to happen and which can be expressed in the future-related statement. These can be in the form of other events with their respective dates, modality or even own conditions.

- **Rationale** defines the logic behind why a given future event is expected to happen. It explains the causation behind an upcoming event, providing a background of the event that would support the particular future course of actions, or other.

- **Modality** indicates the certainty level of a future event made by the statement maker and is often expressed using words like *"will", "likely", "might",* etc.

- **Timestamp** denotes when the particular future-related statement has been made. Naturally, the older the statement, the less probable is that it will be still correct.

Note that not all of the above-discussed elements will be stated in future-related opinions or will be known. For example, humans, when forecasting the future, may not mention the date when the event will occur or there might be no particular prerequisites given. However, when available some of these elements can be quite useful for making the final predictions such as the rationale or credibility of the statement producer. Table 2 highlights some of these components through an example.

## 4.2 Individual Predictions

Some works do not resort to any kind of aggregation or summarization but simply determine individual predictions without performing ranking or clustering. These approaches are often probabilistic. Radinsky et al. (2012) leverage causal reasoning to generate possible future occurrences of an event from a given event. Firstly, they collect event pairs from news articles that are related through a causal relationship

which results in a semantically structured causality graph containing 300 million fact nodes and over 1 billion edges. The Pundit algorithm then applies semantic natural language modeling techniques and a vast amount of world knowledge ontologies to generate an abstraction tree that is used to predict causality between unseen events. Radinsky & Horvitz (2013) undertakes a probabilistic approach by building an inferential model of the form $P(ev_j(\tau + \Delta)|ev_i(\tau))$ for a future event $ev_j$ at time $\tau + \Delta$ and past event $ev_i$ happening at time $\tau$. This approach extracts storylines using topic tracking and detection algorithms, clusters similar texts together, enriches these storylines with information extracted from Web knowledge sources, estimates the above probability, and builds a real-time classifier that indicates the likelihood of a forthcoming event, following a past sequence of events. Hu et al. (2017) develops an end-to-end model called Context-aware Hierarchical Long Short-Term Memory (CH-LSTM) for future sub-event prediction, even if they do not exist in the training data. The model does this by encoding the word sequence for each subevent, the temporal order of sub-events, and the topic of past sub-events for generating context-aware predictions. The evaluation is based on the actual occurrence of the target event and the mean time between the prediction and occurrence.

### 4.3 Clustering

Clustering indicates grouping together related content, in this case, to summarize or forecast future events. Jatowt & Au Yeung (2011) undertook a mixture-model-based clustering approach to estimate the probability of a future event by grouping related events and topics from a large collection of text documents. When a user queries a topic, a cluster is created based on the related events' textual and temporal similarity and is associated with a probability distribution over time. Jatowt et al. (2009) clusters documents containing future temporal expressions based on similar content and time and selects the best clusters to summarize probable future events. A more recent approach by Regev et al. (2024) uses BERTopic [22] to extract topics from document embeddings and clusters these documents topic-wise to eventually summarize a future-event.

### 4.4 Ranking

Ranking future predictions obtained from Web content is useful for summarizing the top results or plotting the predictions in a timeline. Kanhabua et al. (2011) focuses on automatically generating queries from a news story being read by the user, retrieves all predictions related to that news article, ranks them by the degree of their relevance (future information) and finally returns to the user. The ranking algorithm is built on a support vector machine framework, based on features like term, entity, topic and temporal similarity which captures the parity between the generated query and a news prediction.

### 4.5 LLM based predictions

With all the buzz about LLMs, it is now imperative to ponder upon the effectiveness of LLMs in forecasting. With this intention, Schoenegger & Park (2023) evaluated GPT-4's (Achiam et al., 2023) capability of future prediction. It was observed that GPT-4's forecasts are significantly less accurate compared to the median human-crowd forecasts. The authors apprehend that a potential reason for the underperformance is that in making real-world forecasts, the actual answers are unknown, unlike in other benchmark tasks where the strong performance of an LLM could be partially due to the answers being memorized from the training data. Again, in contrast to humans, GPT-4 is not aware of current affairs due to its training cut-off date. Further, Pernegger (2024) investigated the performance of BERT (Devlin et al., 2019), RoBERTa (Liu et al., 2019) and GLM (Du et al., 2022)to conclude that zero-shot learning didn't improve the language models' forecasting abilities and that retrieval-augmented generation (RAG) offers room for substantial improvement. Recently Zhang & Ning (2024) formulated the problem of multi-event forecasting as predicting occurrences of multiple unique relations in a temporal knowledge graph (TKG) for a future day. They design a prompt template such that it does not exceed the maximum permitted token limit for open-source LLMs while including as much historical information as possible. Further, they utilize the pre-trained LLM, RoBERTa-large to encode this prompt and employ a self-attention-based prediction head to handle the output embeddings and

---

[22]https://maartengr.github.io/BERTopic/index.html

predict the relation indicating the future event occurrence. It would also be interesting to investigate the integration of the wisdom of the crowd with LLM forecasts.

### 4.6 Analysis

Some approaches do not predict individual events but show the statistics of future related content in their datasets such as frequency, average sentiment, representative words, etc. Jatowt et al. (2015) analyze a long collection of tweets to quantify temporal attention and related temporal characteristics expressed by users for judging future actions by representing time mentions as a uniform probability distribution over a time duration. Jatowt et al. (2013) conducts an exploratory analysis of future-related information and studies their sentiment degrees in three different languages, namely, English, Japanese and Polish.

Some of the works predict attributes of a future event like date, time, location, etc. instead of what the whole event would look like. Radinsky & Horvitz (2013) predicts numerical attributes like the number of deaths given an event concerning death is predicted by undertaking a binning approach and learning predictors that estimate a probability. Tops et al. (2013) undertakes a classifier-based approach to estimate when an event mentioned in a stream of Twitter micro texts is about to happen, by mapping unseen tweets onto discrete time segments (mainly into broad categories: 'before', 'during', and 'after'). They validate the fact that it is comparatively difficult to predict the time of an event when it is far away, both for a classifier as well as for humans. Hürriyetoğlu et al. (2013) extends this approach for providing hourly forecasts by mapping clusters of tweets to time-to-event. Jin et al. (2020) poses the problem of future event forecasting as a restricted-domain multiple-choice question-answering task on large-scale unstructured data. Time is constrained such that the answer is not present with certainty in the available text, rather temporal understanding and commonsense are required to forecast it.

## 5 Prediction Visualization

Visualizing predictive insights aids the interpretability of results obtained from complex models. Time-series graphs, cluster mapping, word clouds and interactive timelines are some common ways of displaying results to users. In this section, we explore the various visualization tools and their applications.

The Time Explorer built as a part of (Von Krogh et al., 2004) portrays future references in news articles on a timeline. Matthews et al. (2010) aims to improve the visualization capabilities of timelines derived from *Simile Timeline*[23], *Google Trends*[24] and *Google Timeline*[25]. The timeline is split into a "Trend Graph" which shows the change in the frequency of documents containing a query over $n$ years and a "Topic Timeline" which displays the titles of the top-ranked news articles. Jatowt et al. (2009) builds a query-dependent system that generates graph-based visual summaries of the most probable future outcomes of entities such as a person, place, organization, event, etc. This is achieved by analyzing future temporal expressions, detecting periodical patterns in historical documents, clustering these documents based on time and content similarity and finally selecting the best clusters for summarization. Jatowt et al. (2013) analyzes the sentiment degrees of time-referenced future-related information and the amount of information in each dataset across three languages and arranges it on a timeline. While most of the previous works use graphs and static timelines, advancement in future-event prediction visualization has led to a surge in interactive timelines for better user experience. Jatowt et al. (2015) builds an interactive system that helps in collectively visualizing both historical and future perspectives of microblogging and temporal patterns in tweets. They claim that by observing the visualization graphs, researchers can improve the feature engineering for classifiers in the future. Regev et al. (2024) designs a system[26] that takes a user query as input, downloads and preprocesses a large volume of news articles concerned with the query, classifies them to belong to either future-related or non-future-related, clusters them topic-wise and finally displays the future-related content on an interactive

---

[23]http://www.simile-widgets.org/timeline/
[24]http://www.google.com/trends
[25]http://www.google.com/
[26]https://chronicle2050.regevson.com/

timeline. A data point on the timeline represents a ranked topic-wise collection of sentences belonging to the same date.

## 6 Challenges and Future Directions

The survey highlights several old and new approaches undertaken to predict the future by aggregating the crowd's opinion. While problem-solving techniques have evolved over time, a comparatively under-explored domain as this faces numerous challenges. In this section, we identify the potential improvements that would aid researchers in tackling the existing impediments.

- **Multilingual Datasets**: Building datasets across languages from different regions will enable incorporating local context and lead to accurate forecasts. Besides, augmenting existing forecasting models to adapt to multilingual settings would save a huge amount of time and resources.

- **Domain-specific datasets**: So far, datasets have been curated from news collections, web, or social media which provides generalized content. Collecting domain-specific data will add knowledge from niche areas like weather, agriculture, business, and health and help in generating real-time forecasts.

- **Evaluation and Metrics**: Lack of benchmarking in the forecasting community has prevented direct comparison of state-of-the-art models for various categories of forecasting tasks. Besides, there is a need to formulate context-aware metrics which will help in better evaluation.

- **LLMs for forecasting**: GPT-4 (Lavie & Agarwal, 2007) is seen to severely underperform in real-world-forecasting, event if supplied with background information about an event (Yuan et al., 2023). In the future, LLMs' forecasting capabilities can be improved by accessing real-time information (RAG) from the internet for mitigating knowledge cutoff, integrating human-in-the-loop, adapting chain-of-thought prompting, or other prompt engineering techniques.

- **Integrating additional aspects**: Aggregating wisdom of the crows for future event forecasting entails many challenges such as resolving conflict of opinions, retrieving only relevant information and summarizing them in the form of a prediction and its expected date. It is important to consider other dimensions of future-related statements besides just event description as shown in Figure 3. For example, one could measure the confidence level of the predictions as they are based on human assumptions and verdicts which might or might not be exactly accurate.

- **Joint application of Wisdom of the Crowd and other forecasting methods**: Combining public opinion analysis and other forecasting approaches such as time series based forecasting is promising.

## 7 Conclusion

Future forecasting is inherently challenging but, at the same time, it is a very common activity that humans do on a daily basis both in their private and professional lives. It is then natural to try to aggregate individual forecasts of many people using computational approaches. In this paper, we survey the techniques, challenges, data sources and promising avenues of research in future event forecasting using the wisdom of the crowd concept. Our work details the data sources and types, methodologies, evaluation techniques and future directions. We also propose a novel data model (Figure 3) that decomposes future-related statements into elements that carry potential for effective aggregation and subsequently for successful FEP. Our survey could be useful for scholars trying to research this challenging task or practitioners who want to develop intelligent systems for real-time applications.

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
