# OpenReview forum: "Wisdom of the Crowds for Forecasting: Future Event Pre- diction Based on Summarizing Text-based Forecasts"
_TMLR — Rejected by TMLR_

### Review · Reviewer_WC6g · 2024-10-07

**Summary Of Contributions:**

The authors take effort in reviewing and summarizing the literature in the field of Crowd Wisdom-based Future Event Prediction.

**Audience:**

Yes

**Claims And Evidence:**

No

**Requested Changes:**

Significant revisions are needed for this paper to meet the standards of a proper survey. **Below are some suggested directions, but addressing these alone may be not sufficient**:
1. Add a formal description of the studied task.
2. Restructure the paper and rewrite the section titles. Consider presenting the structure visually using a figure, tree, or table.
3. Include additional sections or parts that provide more insightful information, possibly supported by experimental results.
4. Remove filler content to improve clarity and conciseness.

**Strengths And Weaknesses:**

**Strength**: as the contribution.

**Weaknesses**: The survey paper is poorly written, lacks organization, and fails to provide the valuable insights a good survey should deliver. Below are the key issues:
1. **Unprofessional methodology to conduct the survey**. The method used to conduct the survey seems amateurish. The authors mention that they collected all relevant literature through keyword searches on platforms like Google Scholar, which is highly unprofessional in my opinion. $\textcolor{red}{\text{A proper survey should be written by experts who can effortlessly identify key representative works in the field without relying on internet searches}}$, with the rest of the literature expanding upon these foundational studies.
2. **Lack of formal task definition**. The paper does not provide a formal introduction or definition of "Crowd Wisdom-based Future Event Prediction" at the beginning. This omission made it difficult to clearly understand the research problem and how it was being approached throughout the paper.
3. **Lack of well organization**. The structure of the paper is hard to follow, and the section titles are poorly designed. For instance:, 1) Including the related work and publication collections in the introduction is distracting. 2) In section 3, the organization is unclear: titles like "*Based on Time*," "*Based on Patterns*," and "*Based on LLMs*" are inconsistent. The first two subsections focus on aspects of the query questions themselves, while the third abruptly shifts focus to a technique (LLMs), which feels discordant.
4. **Lack of valuable information**: The paper merely describes existing work without offering any insightful analysis. It fails to convey important messages that readers are interested to know, such as which datasets are more comprehensive, which methods are superior, and what common patterns exist across the methods (i.e., a high-level comparison and summary).
5. **Too much filler content**. Many sections of the paper contain unnecessary filler language. For example, in Section 3.3, the following description is overly general and lacks any concrete details on how relevant work utilizing LLMs:
> LLMs have mastered themselves in natural language understanding (contextual, semantic, etc.) and generation. Since LLMs are trained on huge text corpora, they do not need to explicitly collect any data as in all the previous approaches. This information aids an LLM to capture a variety of opinions including those on the future from news articles, web pages and social media, on any specified topic provided by the user. In addition, LLMs can be used as a summarization tool for scrapped web pages. In all of these cases, special prompts need to be designed to elucidate information about the future. A study by Yuan et al. (2023) shows that directly prompting ChatGPT (Ouyang et al., 2022) to identify future-related content leads to suboptimal accuracy. Hence, LLMs provide innumerable avenues for future event extraction.

---

> ### Author Response · Authors · 2024-11-02
> **We have tried to address the feedbacks individually and modified our paper based on the suggestions.**
>
> Thank you so much for the detailed review and feedback. Here are our responses to the issues raised.
>
> 1. “Unprofessional methodology to conduct the survey…”
>
> - Nowadays the Internet houses the most updated research, whether they are published in a conference, a journal or simply on Arxiv. We believe, through extensive searches on platforms like Google Scholar, Semantic Scholar, IEEE explore, ACM Digital Library, Conference webpages, we have been able to collect an almost exhaustive list of works relevant to or representative of the task of future-related summarization.
>
> 2. “Last of formal task definition…”
>
> - We define the concept of “Wisdom of the Crowd for Future Event Forecasting” from the fourth line of the second paragraph under the Introduction section. We keep it crisp and short in the beginning as we discuss many aspects of it throughout the paper. However, we understand that providing a concrete task definition in the beginning is essential for proper readability of the paper. In the modified paper, we have dedicated an entire paragraph in the Introduction section towards explaining the concept of wisdom of the crown for future event forecasting.
>
> 3. “Lack of well organization…”
>
> - We have provided some mentions for related work in Introduction to build the context of our work and maintain a coherent flow of concepts. Based on the suggestions received we have now reformulated the Introduction section to provide a broader overview of the task and move the related works to a different section. We have also provided more intuitive titles to the sub-sections under Section 3 and altered the content organization to a more meaningful pattern.
>
> 4. “Lack of valuable information…”
>
> - We discuss existing works in our paper at various levels, starting from the variety of data used being available online for the task, the languages they are in, the type of contents one can choose and also the size of the datasets available. We have summarized the overall workflow for the wisdom of the crowd summarization for event forecasting by discussing future-related information extraction methodologies in Section 3, event forecasting techniques in Section 4 and the various challenges and promising future works in this area in Section 6. We do not comment on the superiority of any particular method as the underlying problem statements are not always comparable. Rather, we identify important attributes which might improve the task solving. We also propose a data model suggesting possible implementation, supported by examples. We have mentioned about our future work related to this data model in the Conclusion section of the updated paper.
>
>
>
> 5. “Too much filler content…”
>
> - The paragraph about LLMs has been elaborated to promote research in that direction. LLMs have been utilized to a large extent in a lot of tasks. But we have observed that existing works on the wisdom of the crowd method for future event prediction have not used LLMs in their full potential. We have tried to gather all the relevant works that do leverage LLMs for the task and discovered that LLMs are indeed under-explored in this domain. We will add any new work in the modified paper that has been made available from the time we had submitted this survey.
>
> 6. “Requested changes…”
>
> - We very much appreciate the feedback provided. We do agree upon providing a more detailed definition of the task towards the beginning of the paper for the readers to understand the problem better. We have also restructured certain sections of the paper, especially Section 3 for better flow of the paper. Since the paper is meant to be a survey, we do not provide any experimental results at this stage. In future work however, we intend to employ our proposed data model and obtain experimental inferences.  We have mentioned our future work and plans on this data model in the Conclusion section of the modified paper.

---

> ### Comment · Reviewer_WC6g · 2024-11-03
> **Reply to the authors**
>
> Thank the authors for their response.
>
> When I mentioned a "lack of formal task definition," I was referring to the absence of a formalized definition that includes mathematical representations. I hope this clarifies my point.
>
> It is not incorrect at all to rely on online resources to gather relevant literature, but it feels odd to highlight this as the only approach. Typically, survey papers do not elaborate on literature collection methods, as readers assume the authors are experts familiar with the field. Addressing it gives an unintended impression of inexperience. I suggest considering the removal of this part.
>
> > "Since the paper is meant to be a survey, we do not provide any experimental results at this stage,"
>
> I’d like to note that while experimental results are not always required in survey papers, they can certainly add value. The authors could reuse experimental results from their previous papers or conduct a qualitative comparison instead of a quantitative one if they prefer not to run any new experiments. The absence of experimental statistics reduces the impact of this survey.

---

### Review · Reviewer_SPa3 · 2024-10-18

**Summary Of Contributions:**

The paper presents a survey on the methods and frameworks for predicting future events by leveraging crowd wisdom (FEP-CW). It provides a review of current research on future event prediction based on aggregating individual forecasts, which addresses the limitations of time-series forecasting and simulations in handling complex, global events. The authors present existing datasets, outline prediction methods (including clustering, ranking, and LLM-based predictions), and propose a data model that decomposes future-related statements into atomic components useful for event forecasting. The paper also summarizes the shortcomings of current approaches.

**Audience:**

Yes

**Claims And Evidence:**

Yes

**Requested Changes:**

The writing is not very clear. Authors should improve the clarity of writing by providing a clear roadmap of the paper’s structure, main contributions, and how this survey differs from other related work. This should be done within the Introduction section.

Include experiments or case studies demonstrating how the proposed data model can be used to aggregate individual forecasts and generate predictions.


Improve presentation and informativeness of figures.

**Strengths And Weaknesses:**

# Strength

The paper provides a survey of the literature and datasets related to FEP-CW.

Authors propose a data model, which breaks down future-related statements into atomic components, and offer a framework for organizing and analyzing forecasts from multiple sources.

It includes a thoughtful discussion of LLMs for forecasting and evaluates their limitations, which is timely given the rise of LLMs.

# Weaknesses

I am not fully convinced that the problem (i.e. Future Event Prediction on Crowd Wisdom) is significant enough for a survey paper. It’s unclear if the problem presents unique challenges or has notable real-world impact.

While the paper provides a theoretical framework, there is a lack of concrete experiments to demonstrate the efficacy of the proposed data model in real-world scenarios.

The section on LLM-based predictions could benefit from a more in-depth analysis of the specific limitations of current models like GPT-4 and potential improvements.

The justification for the “Data Model of Future-related Statements” is weak; no evidence is provided to support its usefulness.

Section 1.2 on Publication Collection seems overly detailed and doesn’t contribute much to the paper’s overall purpose.

Figures 1, 2, and 3 are not very informative and primarily show simple logic; the font sizes are also too small.

The formatting at the beginning of Sections 3 is problematic (when authors list the options and examples).

---

> ### Author Response · Authors · 2024-11-02
> **We have responded to the individual comments and based on the reviews, made changes to our paper.**
>
> Thank you so much for your valuable feedback. Here are our responses:
>
> 1. “...problem is significant enough…”
>
> - The underlying motivation for this survey, which is future event prediction, is an essential requirement across multiple industries and is a common activity of humans who continuously try to gather and aggregate relevant data in order to infer likely course of future actions. In the first two paragraphs of the Introduction section and the introductory paragraph of Section 4, we have tried to explain the same. However, based on the feedback, we have strengthened the motivation behind this work in the Introduction section and also added an elaborate task definition.
>
> 2. “...section on LLM-based predictions…”
>
> - Only few recent works have leveraged LLMs for future event forecasting. We have discussed the few works that do employ LLMs. For similar reasons, we encourage more use of LLMs for future works in the fourth point of Section 6. We would add any new work in the modified paper that has been made available from the time we had submitted this survey.
>
>
> 3. “...justification for the data model…”
>
> - Since the paper is a survey, mostly based on existing works, we do not provide any experimental results. In the future work, we intend to employ our proposed data model and obtain experimental inferences.  We have mentioned about our future work on this data model in the Conclusion section of the modified paper.
>
>
>
> 4. “Section 1.2 on publication collection…”
> - We included this section to provide a clarification on our exhaustive collection of existing works. However, based on the feedback, we have modified this section into a more concise version and elaborated more on the other sections we were asked to.
>
> 5. “Figure 1,2 and 3 are not very informative …”
>
> - Figure 1 has been included to convey the growth of research in the area of wisdom of the crowd method for future event forecasting. Figure 2 highlights the general workflow for the task and it is essential towards understanding the various requirements to achieve future event predictions. Figure 3 is a suggestive data model which we propose according to our analysis of existing works. We identify key attributes for the task. In future work, we aim to conduct experiments using this data model and publish inferences. Based on feedback, we have added more informative figures and have also corrected the font sizes to make the paper easily readable.
>
> 6. “The formatting at the beginning of Section3…”
>
> - Based on the reviews received, we have restructured Section 3, we changed the titles and altered the contents to make the paper more comprehensible and informative.
>
> 7. “Requested changes…”
>
> - Based on the comments, we have restructured the contents, made several additions and re-formatted/added figures for better readability. Since our paper is a survey paper, we limit our scope to discussing existing works. In the future, we aim to conduct experiments on our proposed data model and publish results and case studies, about which we have mentioned in the Conclusion section.

---

### Review · Reviewer_hGxB · 2024-10-19

**Summary Of Contributions:**

1. This survey mainly focused on future event prediction with crowd wisdom (FEP-CW), collected related works in the last 20 years, and discussed the current research situation, as well as some future directions.
2. This paper is the first survey outlining methods that utilize the “Wisdom of the crowd” to predict future events. It differs from previous surveys that were either too mathematical or application-oriented. Common datasets, concepts, and methods are discussed in an organized way.
3. The authors propose a data model prototype, where they divide a future-related text statement into multiple components, including predicted future event, credibility, conditions, rationale, timestamp, etc.

**Audience:**

Yes

**Claims And Evidence:**

Yes

**Requested Changes:**

Please address the concerns in the weaknesses part.

**Strengths And Weaknesses:**

**Strengths**
1. This paper collected and summarized existing works concerning data sources, information-extracting methodology, and prediction methods. In each section, several related works are discussed and compared with each, with a table or figure illustrating the idea.
2. What’s good is that this paper also discussed LLM-based prediction and RAG-based methods. Though it’s just a brief introduction, I feel glad that some frontier methods are discussed in this paper.
3. The paper also discussed some future directions and challenges, which might be insightful for people in the future.

**Weaknesses**
1. While this paper is comprehensive in terms of the paper discussed, I found most works are mainly compared in overwhelming paragraphs and lack a clear and direct comparison between them in the form of a table or Figure, which is more memorizable.
For example, in section 2, although there is a table 1 summarizing different datasets, the differences between these datasets are not clearly shown. The “Contents“ description is too vague. It’s better to provide some additional information that really distinguishes these datasets, such as data format, dataset motivation, application scenario, etc. The same issue applies to the section 3, where not a single table or figure is used to compare different extraction methods.
2. The concept of the prediction via crowd wisdom and the general idea seems not well introduced. It took me a while to understand the difference between the prediction via crowd wisdom and the prediction via previous records. I suggest the authors should add a comparison in Figure 2 that compares the general workflow of these 2 methods. Or adding some formal definitions of the prediction via crowd wisdom will help people better understand the concept.
3. I found that section 6 seems to only discuss the future directions and ignores the challenges. Maybe it’s better to also summarize some key challenges instead of mixing them with the future directions?

---

> ### Author Response · Authors · 2024-11-02
> **We have made changes in the paper as suggested in the feedbacks. Each individual review has been been addressed in the Comment**
>
> Thank you so much for the comments and review. Here are our responses to address the same:
>
> 1. “While this paper is comprehensive…”
>
> - We have tried to break down the problem into different parts and discuss these parts in detail in sections across the paper. To start with, we provide Figure 2, which gives a general workflow of the problem and correlates to the subsequent sections. We tried to gather every important information about the available data and sources and tried to accumulate them in Table 1 as well as throughout Section 2. The data format of the structured datasets has been discussed extensively in Section 2. Based on the reviews received, we have restructured Section 3, changed the titles, added more tables and figures and altered the contents to make the paper more comprehensible and informative.
>
> 3.  “The concept of the prediction…”
>
> - We define the concept of “Wisdom of the Crowd for Future Event Forecasting” from the fourth line of the second paragraph under the Introduction section. We keep it crisp and short in the beginning as we discuss many aspects of it throughout the paper. However, based on the suggestions received we have reformulated the Introduction section to provide a broader overview of the task and moved the related works to a different section.
>
> 4. “I found the section 6…”
>
> - Keeping in mind the suggestions received, we made a separate section of Challenges and Future Work to explain each of them in further details.

---

### Decision · Action_Editor_wqY9 · 2024-11-26

**Recommendation:** Reject

**Comment:**

Though I know that the authors have made some changes in response to the reviewer's comments, it seems to me that the paper requires a larger revision to create a larger-picture view of the challenges and key insights related to the problem that satisfy the expectations of a TMLR submission.

**Audience:**

The reviewers generally agreed that the topic of the paper is of potential interest to the TMLR audience. However, overall, the reviewers felt that the survey failed to provide sufficient additional well-supported insight beyond the summary of the papers considered. Recall that, in the call for submissions, TMLR invites "surveys that draw new connections, highlight trends, and suggest new problems in an area." Some
of the specific concerns included the difficulty in direct comparison of the approaches discussed, lack of in-depth analysis, and missing discussion of open challenges in the field.

**Claims And Evidence:**

The paper does not set out to make firm scientific or theoretical claims, instead focusing on collecting and categorizing existing work under the umbrella of future event prediction using crowd wisdom (FEP-CW).

The paper does introduce a framework for decomposing a prediction into component pieces, which is speculative in nature is somewhat disconnected from the survey. The text state that it "might be useful" but reviewers felt that even this mild claim was insufficiently supported as the framework is described without analysis or evaluation.

**Resubmission Of Major Revision:**

The authors may consider submitting a major revision at a later time.